# Nutritional and Morphofunctional Assessment of Post-ICU Patients with COVID-19 at Hospital Discharge: NutriEcoMuscle Study

**DOI:** 10.3390/nu16060886

**Published:** 2024-03-19

**Authors:** Clara Joaquín, Irene Bretón, María Julia Ocón Bretón, Rosa Burgos, Diego Bellido, Pilar Matía-Martín, Miguel Ángel Martínez Olmos, Ana Zugasti, María Riestra, Francisco Botella, José M. García Almeida

**Affiliations:** 1Department of Endocrinology and Nutrition, Hospital Germans Trias i Pujol, Carretera del Canyet, S/N, 08916 Badalona, Spain; 2Department of Endocrinology and Nutrition, Hospital General Universitario Gregorio Marañón, C/Dr. Esquerdo, 46, 28007 Madrid, Spain; irenebreton@gmail.com; 3Department of Endocrinology and Nutrition, Hospital Clínico Universitario Lozano Blesa, C/San Juan Bosco, 15, 50009 Zaragoza, Spain; mjocon@salud.aragon.es; 4Nutritional Support Unit, Hospital Universitario Vall d’Hebron, Pg. de la Vall d’Hebron, 119, Horta-Guinardó, 08035 Barcelona, Spain; rosa.burgos@vallhebron.cat; 5Department of Endocrinology and Nutrition, Complejo Hospitalario Universitario de Ferrol (A Coruña), Av. da Residencia, S/N, 15405 Ferrol, Spain; diegobellido@gmail.com; 6Department of Endocrinology and Nutrition, Hospital Clínico Universitario San Carlos, 28040 Madrid, Spain; pilar.matia@gmail.com; 7Facultad de Medicina, Universidad Complutense de Madrid, 28040 Madrid, Spain; 8Instituto de Investigación Sanitaria San Carlos (IdISSC), 28040 Madrid, Spain; 9Department of Endocrinology and Nutrition, Hospital General Universitari, 15706 Santiago de Compostela, Spain; miguel.angel.martinez.olmos@sergas.es; 10CIBERObn, Instituto Nacional de Salud Carlos III, 28029 Madrid, Spain; 11Grupo de Investigación de Endocrinología Molecular, Instituto de Investigación Sanitaria de Santiago de Compostela-IDIS, 15706 Santiago de Compostela, Spain; 12Section of Clinical Nutrition and Dietetics, Complejo Hospitalario Universitario de Navarra, C/de Irunlarrea, 3, 31008 Pamplona, Spain; azugas@hotmail.com; 13Department of Endocrinology and Nutrition, Hospital Universitario de Cabueñes, Los Prados, 395, 33394 Gijón, Spain; mriestra.fernandez@gmail.com; 14Spanish Society of Endocrinology and Nutrition, C/Villalar, 7, 28001 Madrid, Spain; botellaromero@gmail.com; 15Department of Endocrinology and Nutrition, Hospital Virgen de la Victoria, 29010 Málaga, Spain; jgarciaalmeida@gmail.com; 16Hospital Quiron salud, UMA, 29010 Málaga, Spain; 17Instituto de Investigación Biomédica de Málaga (IBIMA), 29010 Málaga, Spain

**Keywords:** SARS-CoV-2, COVID-19, intensive care units, nutritional status, body composition, functional status, malnutrition, nutrition assessment, bioelectrical impedance analysis, nutritional ultrasonography

## Abstract

This paper presents baseline results from the NutriEcoMuscle study, a multicenter observational study conducted in Spain which focused on changes in nutritional status, body composition, and functionality in post-intensive care unit (ICU) COVID-19 patients following a nutritional intervention. Assessments at hospital discharge included Subjective Global Assessment (SGA), Global Leadership Initiative on Malnutrition (GLIM) criteria, the Barthel index, handgrip strength (HGS) and the Timed Up-and-Go test, bioelectrical impedance analysis (BIA), and nutritional ultrasound (US). The study involved 96 patients (71.9% male, mean age 58.8 years, mean BMI 28.8 kg/m^2^, 36.5% obese). All patients were malnourished at discharge according to GLIM and SGA. Functional status declined from admission up to hospital discharge. A total of 33.3% of patients had a low fat-free mass index (FFMI) and 29.5% had a low phase angle (PhA). Myosteatosis was observed in 83.7% of the population. There was a positive correlation between rectus femoris cross-sectional area, PhA, FFMI, and HGS. In conclusion, post-critically ill COVID-19 patients commonly suffer from malnutrition and reduced muscle mass, causing a loss of independence at hospital discharge. BIA and US could be valuable tools for assessing body composition in these patients. The NutriEcoMuscle study highlights the need for a thorough nutritional and morphofunctional status assessment of post-ICU patients.

## 1. Introduction

Coronavirus 2019 disease (COVID-19) is a respiratory infectious disease caused by the severe acute respiratory syndrome coronavirus 2 (SARS-CoV-2) [1]. The start of the COVID-19 pandemic had devastating health effects. It led to high admission rates (32%) to intensive care units (ICUs) and high mortality rates (39%) for ICU patients [2]. Severely ill patients often required prolonged ICU admission that can be extended to an average of 53 days, leading to nutritional and physical problems which required special attention [3,4]. Poor physical condition upon discharge is also associated with higher rates of readmission and mortality [5]. Several studies have shed some light on the nutritional and functional status of COVID-19 patients during their discharge from the intensive care unit (ICU). However, it is still unclear what the incidence of malnutrition among COVID-19 survivors is and what the long-term health consequences may be. Furthermore, COVID-19 may serve as a model of a severe disease with significant functional and nutritional repercussions [6] that can be applied to other pathologies that cause Acute Respiratory Distress Syndrome (ARDS).

Nutritional ultrasound^®^ is a new technique for the evaluation of body composition. It provides information about muscle mass and body fat mass in a noninvasive and portable way. This is an emerging and inexpensive method that uses linear, broadband, multifrequency probes with a depth field of 20–100 mm to evaluate the musculoskeletal apparatus. It can quantify muscle and adipose tissue in various body locations. Changes in muscle echogenicity can also provide information on functional muscle status [7,8].

The NutriEcoMuscle study was an ambispective multicenter study conducted in Spain. Its objective was to evaluate the body composition changes in patients discharged from the ICU after being treated for COVID-19. The study involved an intervention that included oral nutritional supplementation with 100% serum lactoprotein enriched with leucine and vitamin D combined with motor rehabilitation.

The objective of the current work was to describe the nutritional and morphofunctional status of patients admitted to the ICU due to COVID-19 at the time of their hospital discharge, which is the baseline of the NutriecoMuscle study. These data may serve as a model of severe disease and can be extrapolated to other pathologies that cause ARDS. Additionally, we aimed to analyze the differences in morphofunctional parameters based on the diagnosis and severity of malnutrition. Our secondary objective included the evaluation of the accuracy of ultrasound (US) measurements compared to bioelectrical impedance analysis (BIA), which is a validated technique for body composition assessment.

## 2. Materials and Methods

This was an ambispective, observational, multicenter study carried out from March 2021 to January 2022 in ten hospitals in Spain.

This project was approved by the Ethics Committee of the Hospital Universitari Germans Trias i Pujol (Badalona, Spain) (code PI-20-321). The procedures and materials complied with the principles of the Declaration of Helsinki and with regulations on data protection and research in Spain (Ley Orgánica 3/2018).

### 2.1. Participants

This study included adult patients aged between 20 and 75 years who had been admitted to the ICU because of severe COVID-19 and had a hospital stay lasting over 72 h. A positive SARS-CoV-2 real-time fluorescence polymerase chain reaction (RT-PCR) test confirmed the COVID-19 diagnosis. Management of COVID-19 patients in the ICU was carried out following local clinical protocols. The exclusion criteria were as follows: pregnancy, patients with standing difficulties, patients with amputations, patients who had a Barthel index (BI) score less than 60 (indicating severe dependency) before admission, patients with a previous body mass index (BMI) greater than 50 kg/m^2^, and patients who did not provide consent.

### 2.2. Variables

Medical records provided the patients’ demographic and clinical characteristics: age, BI before hospital admission, length of hospital and ICU stay, comorbidities, Sequential Organ Failure Assessment Score (SOFA), and need for orotracheal intubation.

Nutritional status was assessed using the Subjective Global Assessment (SGA) questionnaire [9], and malnutrition was diagnosed according to the Global Leadership Initiative on Malnutrition (GLIM) criteria [10]. The diagnosis of malnutrition, according to the GLIM criteria, is based on the presence of at least one of three phenotypic criteria combined with one of two etiological components. The phenotypic criteria are (i) non-volitional weight loss > 5% within the last six months or >10% over a six-month period; (ii) BMI < 18.5 kg/m^2^ at age < 70 years or <20 kg/m^2^ at age ≥ 70 years; and (iii) reduced muscle mass. Reduced muscle mass was assessed according to bioelectrical impedance (BIA) measurements of the fat-free mass index (FFMI) (<17 kg/m^2^ in males and <15 kg/m^2^ in females). Clinicians determined the severity of muscle mass depletion by physical examination [10]. The etiological components of the GLIM criteria are as follows: (iv) reduced food intake (<50% of energy requirements for one week or any reduction for over two weeks) or assimilation and (v) inflammatory status. Reduced food intake was evaluated using quartiles and food assimilation per clinical record. Disease-related inflammation was identified when C-reactive protein (CRP) concentration was >5 mg/L at hospital discharge. BMI was used to assess overweight and obesity status according to accepted international classifications [11]. Handgrip strength was measured on three separate occasions with a Jamar^®^ dynamometer, Asimow Engineering Co., Los Angeles, CA, USA (in the second handle position). This was performed with the patient seated on a chair, with their shoulder and forearm in a neutral position and the elbow at 90 degrees of flexion. The participant performed a maximum grip force for 3 s and rested for 1 min between each repetition. The average of the three measurements was used for the analyses following the recommendations of the American Society of Hand Therapists [12]. The handgrip strength reference values used to diagnose dynapenia were the cut-offs recommended by the EWGSOP2 consensus (<27/16 kg for males/females) [13]. The Timed Up-and-Go (TUG) test was also performed to assess physical performance. The test was considered pathological if the patient needed more than 20 min to complete the test [14].

Different BIA devices were used depending on the hospital where the test was performed: BIA 101 BIVA (Akern, Pontassieve, Italy; www.akern.com); NUTRILAB (Akern, Pontassieve, Italy; www.akern.com); QUADSCAN 4000 (Bodystat, Douglas, Isle of Man, UK; www.bodystat.com); INBODY 770 (Inbody, Seoul, Republic of Korea, www.inbody.com); INBODY S10 (Inbody, Seoul, Republic of Korea, www.inbody.com); SECA 525 (Seca, Catalonia, Spain, www.seca.com/es; TANITA 780 (Tanita, Arlington Heights, IL, USA, www.tanita.com). To perform BIA, according to standard recommendations, patients were not allowed to ingest any liquids two hours before the test. Before taking the measurements, the subject rested in a supine position for 4–5 min while their data (gender, age, weight, and height) were being entered into the device. Then, the test was performed following the manufacturer’s instructions for each type of equipment. The BIA approach allows for the quantification of body composition parameters either through predictive equations as set by the manufacturer or providing the raw resistance (R) and reactance (Xc) to be inserted into specific formulas by the operator. Most formulas have been developed for general or specific populations but not for COVID-19 disease [15,16,17,18]. Total body water, whose estimations for fat-free mass (FFM) are based on the usage of proprietary equations, was calculated only using 50 kHz impedance. The BIA variables recorded in the present study were FFM and phase angle (PhA). Cut-off points were established according to the guidelines for undernutrition (GLIM) [10]. Patients were categorized for low phase angle (<3.95°) following a previous report on COVID-19 patients [19].

To perform a nutritional ultrasound^®^, a Mindray Z7 US (Shenzhen Mindray Bio-Medical Electronics Co., Ltd., Shenzhen, China) was used in all participating centers. Adipose tissue and musculoskeletal areas were evaluated with a 10–12 MHz soft tissue transducer and a multifrequency linear array probe (probe width 40 mm). The rectus femoris (RF) muscle was evaluated with the patient in the supine position and with the transducer placed transversely at the lower 1/3 of the distance between the anterosuperior spine of the pelvis and the upper edge of the patella. Muscle area, circumference, and longitudinal and transverse distance measurements were obtained [7]. Signs of muscle fatty infiltration (myoesteatosis) were also recorded. For the evaluation of abdominal subcutaneous adipose tissue, the transducer was placed at the midpoint between the xiphoid process and the umbilicus and the total (TAT), superficial (SAT), and preperitoneal (PAT) adipose tissues were measured. Images were taken during unforced expiration, in a transverse axis, and with an alignment perpendicularly to the skin. Visceral adipose tissue was determined by measuring the distance between the edge of the parietal peritoneum and the inner face at the junction of the two rectus abdominal muscles. To reduce interobserver variability, all investigators were trained to follow this specific protocol regarding nutritional ultrasound before initiating the study.

### 2.3. Statistical Analysis

Demographic and clinical data are depicted using descriptive statistical indices. Quantitative variables are expressed as mean and standard deviation or median and interquartile range, while frequencies and percentages were calculated for qualitative variables. Statistical differences were compared using the Mann–Whitney, Fisher, and chi-square tests. The degree of agreement between the SGA and the GLIM criteria was assessed by Cohen’s kappa correlation index. Correlations between functional assessment results and body BIA and US composition parameters were performed and analyzed by Spearman’s correlation coefficient. A *p*-value < 0.05 was considered statistically significant. Statistical analyses were performed using SAS v9.3 statistical software (SAS Institute, Cary, NC, USA).

## 3. Results

A total of 96 patients were included in this study. Table 1 displays the demographic and clinical data for the selected sample, classified by gender. Patients were predominantly male; the mean age (SD) was 58.8 (8.5) years, and the mean BMI was 28.8 (5.8) kg/m^2^. No patients had a BMI < 18.5 kg/m^2^. The mean length of hospital stay was 48.2 (37.6) days, with an ICU stay of 28.7 (27.5) days. The mean number of recorded comorbidities was 1.1, and 50% of the population reported 0 to 2 pathologies. Obesity (41.7%) and high blood pressure (HBP) (35.4%) were the most prevalent comorbidities. Upon admission, most of the participants (91.6%) were functionally independent, with a mean score of 99.0 as per the BI assessment. No significant differences were found in population characteristics by gender, except for obesity, where female patients had a significantly higher prevalence than male patients (*p* = 0.0386).

### 3.1. Nutritional Evaluation

Nutritional evaluation based on the SGA at hospital discharge is shown in Table 2. According to the SGA, all patients were malnourished, with 52.1% moderately malnourished (SGA-B) and 47.9% severely malnourished (SGA-C). The mean weight loss during the preceding six months was 11.0 (7.1) kg, representing a loss of 11.6 (6.7)%. The SGA-C group showed significantly greater weight loss than the SGA-B group over the last six months (*p* < 0.0001). Physical examination showed that 82.3% of the patients had subcutaneous fat loss and 83.3% of them muscle mass loss, with the SGA-C group having significantly more severe loss than the SGA-B group for both parameters (*p* < 0.0001 and *p* = 0.0002, respectively).

Based on the GLIM criteria, all patients were malnourished at hospital discharge (45.8% moderately and 54.2% severely malnourished) (Table 3). The total weight loss at six months was significantly higher in the severely malnourished group (*p* < 0.0001), where 94.2% experienced a >10% weight loss within the last six months. Low BMI (<20 kg/m^2^ in patients with <70 years or <22 kg/m^2^ with ≥70 years) was evident in three patients (3.1%), without any differences between moderately or severely malnourished groups. Reduced muscle mass (according to bioelectrical BIA measurements of FFMI) was present in half of the population. Muscle mass reduction was observed in 50.1% of patients, with 31.3% showing mild to moderate and 18.8% severe reduction. None of the patients with moderate malnutrition had severe loss of muscle mass. Reduced dietary intake (or absorption) and inflammation were found in 60.4% and 91.7% of patients, respectively. No significant differences in these parameters were found between the moderately and severely malnourished groups.

### 3.2. Functional Assessment

Before hospital admission, 8.4% of patients were dependent for their needs (BI score < 100). At discharge, the median BI score was 90, and 66.7% of patients had some degree of dependency, but none were totally dependent (Table 4).

The proportion of patients with severe, moderate, and mild dependency increased from admission to discharge (Figure 1).

Table 5 presents the results of the handgrip strength and TUG tests at hospital discharge. A total of 60 patients (62.5%) showed handgrip strength below the values recommended by the EWGSOP2 (<27 kg for men or <16 kg for women). Women had a higher proportion of low handgrip strength than men (77.8% vs. 56.6%), but the differences did not reach statistical significance. The average TUG test score was 20.0 ± 17.3 s, with men performing significantly better than women (mean 16.7 [14.2] s vs. 28.1 [21.5] s; *p* = 0.0004). A total of 26 patients (27.4%) had TUG test scores above the cut-off value of 20 s. Women had a significantly higher proportion of pathological TUG tests than men (44.4% vs. 20.3%, *p* = 0.0224).

### 3.3. Body Composition Assessment

Table 6 provides information regarding BIA and nutritional US measurements at hospital discharge. The women’s group exhibited significantly lower FFMI (*p* = 0.0341) values than the men’s group. Low FFMI (<17 men or <15 kg/m^2^ women) was observed in 33.3% of the population, without differences between men and women. The mean phase angle was 4.5° (1.1) without gender differences. A low phase angle (<3.95°) was observed in 29.5% of the patients.

Nutritional ultrasound measurements revealed values of 0.9 (0.5) cm for preperitoneal adipose tissue, 1.0 (0.6) cm for rectus femoris (RF) thickness, and 3.4 (1.3) cm for rectus femoris cross-sectional area (RF-CSA). The women’s group displayed significantly lower values for RF-CSA (*p* < 0.0001) and RF thickness (*p* < 0.0001) as well as higher subcutaneous abdominal adipose tissue (*p* < 0.0001) when compared with the men’s group. Myosteatosis was observed in 83.7% of the studied population: 100% of the women and 78.1% of the men.

When we analyzed US parameters in patients with or without low FFMI (<17 kg/m^2^ in men or <15 kg/m^2^ women), we observed that men with low FFMI had significantly lower RF-CSA (mean 2.89 [1.21] vs. 4.11 [1.26] cm^2^; *p* < 0.0001), lower RF thickness (mean 0.86 [0.27] vs. 1.24 [0.33] cm; *p* < 0.0001), and lower RF circumference (mean 8.52 [1.09] vs. 9.25 [1.42] cm; *p* = 0.037) than men with average FFMI values. In the women’s group, there were no differences between those with or without low FFMI regarding RF-CSA (mean 2.47 [0.56] vs. 2.56 [0.58] cm^2^; *p* = 0.77), RF thickness (mean 0.79 [0.15] vs. 0.83 [0.15] cm; *p* = 0.70), and RF circumference (mean 7.85 [1.53] vs. 8.31 [1.79] cm; *p* = 0.57).

A positive correlation was observed between RF-CSA and phase angle (rho = 0.51; *p* < 0.0001), FFMI (rho = 0.41; *p* < 0.0001), and handgrip strength (rho = 0.55; *p* < 0.0001) (Figure 2). RF thickness also correlated with FFMI (rho = 0.46; *p* < 0.0001).

## 4. Discussion

This study aimed to analyze the body composition and the nutritional and functional status at hospital discharge of post-ICU COVID-19 patients as a model of severe disease that can be extrapolated to other pathologies that cause ARDS. Additionally, we investigated how both malnutrition and gender impact body composition and functional parameters. Furthermore, we explored the clinical applicability of new tools for body composition measurement, such as nutritional US, compared to classical techniques, such as BIA.

At hospital discharge, all post-ICU COVID patients exhibited varying degrees of malnutrition with substantial and sudden weight loss during their hospital stay. Previous studies have also reported a significant prevalence of malnutrition [20,21,22]. The causes of this weight loss and malnutrition are multifactorial. Acute systemic inflammation, present in patients with severe COVID-19, alters metabolic and hypothalamic pathways, causing anorexia, reduced food intake, increased resting energy expenditure, and accelerated muscle catabolism [23,24]. Acute inflammatory events might elicit chronic neuroinflammatory responses in sensitive individuals, prolonging inflammation and wasting [25]. In addition, illness duration and hospital length of stay are weight loss predictors in multivariate models [26]. This could also explain the results found in the present study, considering that our cohort’s average length of stay was 48 days.

Our study classified patients based on their nutritional status using two different criteria: the SGA and the GLIM criteria. The validity of the GLIM criteria in post-ICU patients has not been extensively researched. Therefore, our study evaluated the validity of the GLIM criteria when diagnosing disease-related malnutrition in post-ICU COVID-19 patients and compared it to the SGA. According to the obtained results, the GLIM criteria were found to be a good indicator of disease-related malnutrition in the studied patients, exhibiting similar results to the SGA regarding the diagnosis of malnutrition. However, there was only a fair agreement between the two methods in identifying different stages of malnutrition. Our results are consistent with the ones observed in a recent study performed in critically ill COVID-19 patients 48 h after ICU admission, where an optimal agreement between the GLIM criteria and the SGA regarding malnutrition diagnosis was observed [27]. Additionally, this study found that malnutrition assessed by the GLIM criteria had a strong association with mortality and longer ICU stay duration [27]. In line with our results, another study assessing hospitalized patients showed a fair agreement between malnutrition stages. The GLIM criteria were found to under-represent overall malnutrition when compared to the SGA. However, the GLIM criteria were more effective in identifying severely malnourished individuals [28]. Using both methods, we could observe that severe malnutrition was associated with a more significant weight and muscle mass loss. Concerning the GLIM criteria, BMI, dietary intake, and levels of inflammation did not succeed in differentiating among the different stages of malnutrition. Only weight loss and severity of muscle mass loss were the signs that could guide the diagnosis. Considering this, body composition assessment seems crucial for the nutritional evaluation of post-ICU patients.

The study data show that the COVID-19 patients admitted to intensive care in this study had a mean age of 58.8 years and were mostly male, which is consistent with other studies [20,21,22]. Overweight and obesity were highly prevalent in our cohort, as reported by other authors as well [29]. Obesity may increase the risk of COVID-19 infection, hospitalization, clinically severe illness, mechanical ventilation, ICU admission, and death [30]. This relationship is considered to be the result of many processes, including the angiotensin-converting enzyme-2 (ACE-2) pathway and increased inflammation caused by adipocytes [31,32]. The high prevalence of obesity in COVID-19 patients, especially in women, makes the diagnosis of malnutrition more complex since most of them have a normal or high BMI. Sex differences in BMI data were also explored during our study, and it was found that a higher percentage of women fell into the obesity category. Studies performed during the COVID-19 pandemic observed that male patients are almost three times as likely to require ICU admission as female patients [33,34]. Therefore, the ICU admission of female patients could be related to the presence of severe COVID-19 worsening factors, including obesity. Obesity is a well-known risk factor of ICU admission for COVID-19 patients [35,36].

Functional decline is common after an ICU stay [37], which is also evident in COVID-19 patients [38,39,40,41]. In our study, more than half of the analyzed population experienced functional decline at hospital discharge, according to the BI scores. Moreover, nearly two-thirds exhibited low handgrip strength, while 27.1% of the patients were at higher risk of falling, according to the TUG test. The high degree of physical dysfunction was consistent with the results of other similar studies [20,21,22]. Several factors may explain the results, including sarcopenic obesity, inflammatory mediators, and insulin resistance [42]. Skeletal muscle atrophy appears after more than 72 h of immobility, even in healthy patients, and is aggravated when patients are confined to bed for lengthy periods [43], as in our cohort. It is worth noting that, according to our study, women had a poorer functional status at hospital discharge compared to men. Specifically, in the TUG test, men performed significantly better than women, as women had a significantly higher proportion of pathological TUG tests compared to men (44.4% vs. 20.3%). Moreover, a higher proportion of women presented lower handgrip strength than men (77.8% vs. 56.6%), although the differences did not reach statistical significance. Women presented with a higher prevalence of myoesteatosis in our study, which is recognized to correlate negatively with strength and mobility [44]. Given the above considerations, it is crucial to recognize the influence of both hospital and ICU stays on the functional status of individuals with ARDS, particularly women. It is necessary to establish and execute protocols that can reduce the impact that hospital and ICU admissions might have.

Concerning body composition assessment, several factors need discussion. When analyzing BIA measurements, FFMI values were very similar between men and women when one would expect them to be lower in women. One explanation for this finding is that, in our cohort, women were more obese than men, and obesity may overestimate relative muscle mass when this is expressed in relation to height, such as in the FFMI. Therefore, experts advise adjusting body composition data to body weight in the recent ESPEN and EASO consensus statements regarding the definition of and diagnostic criteria for sarcopenic obesity [45]. This could explain the fact that although women exhibited worse functionality than men, there were no gender differences in the percentage of low FFMI. Furthermore, it must be noted that when the SGA was performed, 83.3% of the patients exhibited reduced muscle mass, whereas only half of the population demonstrated such reduction with the application of the GLIM criteria, in which a low FFMI was used to evaluate decreased muscle mass. This suggests a potential overestimation of the FFMI, possibly influenced by the elevated proportion of obese patients.

Nutritional ultrasound (muscular and adipose tissue) is an emerging technique in nutritional assessment, which, in recent years, has undergone significant development [7]. Therefore, it is important to determine US’s usefulness as a potential tool for this purpose compared to established techniques, such as BIA. Moreover, apart from measuring muscle morphology parameters, such as length, volume, and area, the assessment of echogenicity provides information on muscle architecture and quality by assessing items such as fat infiltration. In our cohort, we found a good correlation between US parameters such as RFA-CSA, RF thickness, and RF circumference with validated parameters of muscle mass such as FFMI. Moreover, men who were identified by the FFMI as sarcopenic had significantly lower RFA-CSA, RF thickness, and RF circumference. Nevertheless, we could not observe these findings in women, probably due to the low number of women in our study and the higher prevalence of obesity in this group, which could have interfered with FFMI measurements. Other authors have described US as a valuable tool in assessing muscle atrophy in ICU patients with good intra- and interobserver reliability [46,47]. On the other hand, RF-CSA correlated well in the whole cohort with PhA. PhA is a predictor of mortality in diverse clinical conditions and a potentially helpful screening tool for prognosis [48]. In this way, a PhA < 3.95° within the first 72 h after hospital admission has been identified as a significant predictor of mortality risk in COVID-19 patients independent of age, sex, BMI, and comorbidities [19]. In our cohort, practically one-third of the patients had a low PhA (<3.95) at discharge, which could suggest impaired prognosis in this group of patients. Nevertheless, we do not have information about the clinical implications of these low-phase angles at discharge. Moreover, in our study, RF-CSA measured by US correlated positively with handgrip strength, which is a well-known prognostic tool in the general population [49,50], and has been shown to independently and inversely predict poor outcome risk in people with COVID-19-related pneumonia [51]. In 2016, Mueller et al. first proposed sex-adjusted RF-CSA defined by ultrasound as a tool to predict hospital mortality and longer hospital stays in a cohort of surgical ICU patients [52]. Andrade-Junior et al. demonstrated in critically ill COVID-19 patients that a loss of muscle mass occurred in the first ten days of stay in the ICU [53]. They showed a reduction of 30.1% in RF-CSA and of 18.6% in muscle thickness. Umbrello et al. also showed in critically ill COVID-19 patients that a loss of muscle mass occurred in the first seven days of their ICU stay [54]. They revealed a change in RF-CSA of less than 17.9% for survivors, whereas non-survivors exhibited a change of less than 36.3%. These findings suggest that nutritional US could be an important tool for a rapid bedside prognostic evaluation of critical patients after discharge from the ICU. It has to be noted that myoesteatosis was highly prevalent in our population, with a higher prevalence in women. This could explain why the rectus femoris muscle in women was larger than expected and less functional. Regarding adipose tissue assessed by US, women exhibited higher subcutaneous abdominal adipose tissue than men, as reported by other authors as well [55]. Nevertheless, preperitoneal adipose tissue, which is usually more predominant in males, did not differ by gender, probably because females were more obese than males in our study.

In summary, patients with COVID-19 have a high prevalence of obesity, which complicates the diagnosis of malnutrition since body composition techniques such as BIA may not be appropriate to detect sarcopenia in this type of patient. This can be extrapolated to all critically ill patients with obesity. Consequently, incorporating ultrasound as a supplemental tool to BIA could be very valuable for assessing body composition in post-critically ill patients. Moreover, PhA and RF-CSA measurements are novel options for the practical assessment and clinical evaluation of an impaired nutritional status and prognosis among hospitalized COVID-19 patients. They could contribute to enhanced patient care and clinical outcomes. We consider that the evaluation of functional status and body composition assessments with BIA and US as a complementary tool are valuable tools for the clinical management of post-ICU patients and should be included in routine clinical assessment during hospital admission. A comprehensive nutritional and functional assessment within a multidisciplinary team is crucial in these patients to enhance outcomes.

There were several limitations in our study. First, this study included 96 patients; therefore, the sample cannot be considered fully representative of the totality of the COVID-19 patients who have undergone ICU stays. Second, this research was conducted during the fourth and sixth outbreaks of COVID-19 in Spain [56]. Therefore, the SARS-CoV-2 variants that existed at the time may be distinct from those that may exist today or in the future, which is an additional element to consider when interpreting the findings. Another limitation of this study was the use of different BIA devices to assess body composition, depending on the hospital where the test was performed. Nevertheless, to minimize bias, total body water, necessary for estimating fat-free mass (FFM), was calculated only by using 50 kHz impedance. Additionally, BIA muscle mass evaluations in obese patients may have been overestimated, potentially leading to the underdiagnosis of malnutrition in these patients when the GLIM criteria were used. However, one of the strengths of our study is that to minimize interobserver variability with US, specific training was given to researchers regarding nutritional US, and the same US model was used in all hospitals that participated in the study.

## 5. Conclusions

The NutriEcoMuscle study’s preliminary results revealed that all post-critical COVID-19 patients suffered from some degree of malnutrition at hospital discharge, despite a high prevalence of overweight or obesity. Also, a high occurrence of reduced muscle mass, leading to a loss of independence, was detected. Furthermore, this study shows that body composition assessment is crucial for the diagnosis of malnutrition in post-ICU patients, and that muscle and adipose tissue ultrasound could be a very valuable and rapid bedside tool, complementary to BIA, for assessing body composition in these patients. These data may serve as a model of severe disease that can be extrapolated to other pathologies that cause ARDS. Incorporating body composition assessment by using BIA and nutritional ultrasound is feasible in routine clinical practice and should be an integral part of the clinical assessment in hospitalized patients.

## Figures and Tables

**Figure 1 nutrients-16-00886-f001:**
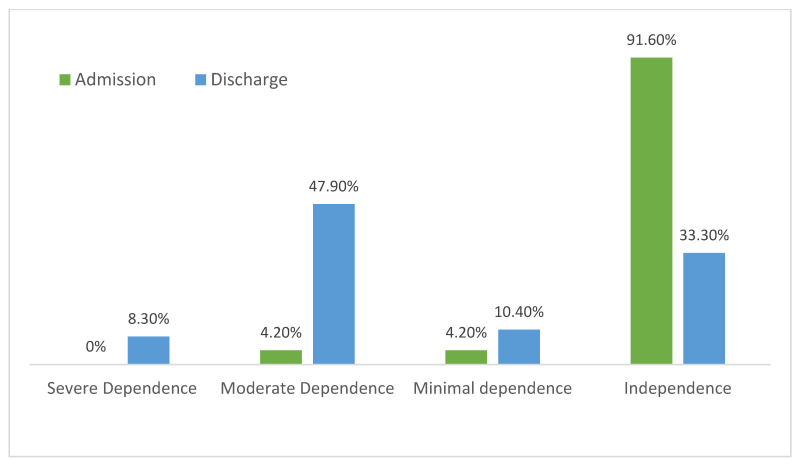
Degree of dependency according to the Barthel index before admission and at hospital discharge.

**Figure 2 nutrients-16-00886-f002:**
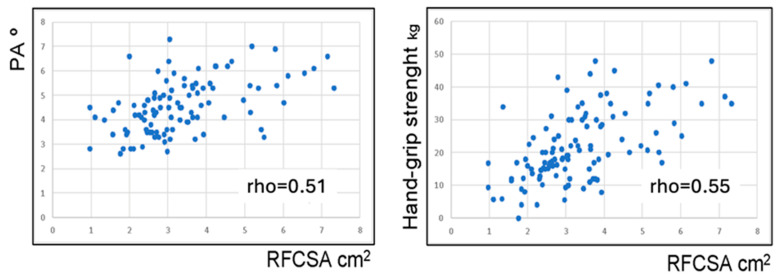
Correlation of RF-CSA with PA and handgrip strength. Abbreviations: PA, phase angle; RF-CSA, rectus femoris cross-sectional area.

**Table 1 nutrients-16-00886-t001:** Characteristics of the sample.

	Total	Men	Women	*p*-Value
n (%)	96 (100.0)	69 (71.9)	27 (28.1)	-
Age, mean (SD), y	58.8 (8.5)	58.5 (8.8)	59.7 (7.7)	NS
Age ≥ 65 years, n (%)	29 (30.2)	20 (29.0)	9 (33.3)	NS
BMI, mean (SD), kg/m^2^	28.8 (5.8)	27.9 (5.1)	30.9 (6.2)	NS
BI scores upon admission	99.0 (4.9)	98.8 (5.6)	99.6 (1.9)	NS
Comorbidities, n (%):				
Obesity	40 (41.7)	24 (34.8)	16 (59.3)	0.0386 *
HBP	34 (35.4)	22 (31.9)	12 (44.4)	NS
Diabetes mellitus	19 (19.8)	13 (18.8)	6 (22.2)	NS
COPD	6 (6.3)	5 (7.2)	1 (3.7)	NS
CKD	2 (2.1)	2 (2.9)	0 (0.0)	NS
CHF	3 (3.1)	3 (4.3)	0 (0.0)	NS
Active oncologic pathology	2 (2.1)	2 (2.9)	0 (0.0)	NS
Length of stay, mean (SD), days	48.2 (37.6)	49.7 (40.9)	44.6 (27.6)	NS
Pre-ICU hospital stay, mean (SD), days	2.3 (3.2)	2.0 (2.4)	3.1 (4.7)	NS
ICU stay, mean (SD), days	28.7 (7.5)	30.3 (29.7)	24.4 (20.4)	NS
ICU characteristics, mean (SD):				
SOFA score	4.1 (2.4)	4.2 (2.7)	4.0 (2.3)	NS
Mechanical ventilation, n (%):	58 (60.4)	40 (58.0)	18 (66.7)	NS
NIMV	5 (5.2)	4 (5.8)	1 (3.7)	NS
HFNC	33 (34.4)	25 (36.2)	8 (29.6)	NS
CRP, median (IQR), mg/dL	2.9 (0.5–9.0)	2.9 (0.4–11.3)	2.9 (0.7–8.4)	NS

Abbreviations: HBP, high blood pressure; BI, Barthel index; CHF, congestive heart failure; CKD, chronic kidney disease; COPD, chronic obstructive pulmonary disease; HFNC, high flow nasal cannula; ICU, intensive care unit; IQR, interquartile range; NIMV, noninvasive motion ventilation; SOFA, Sequential Organ Failure Assessment; CRP, C-reactive protein, BMI, body mass index; NS: not statistically significant. * *p*-value: comparison between BMI < 30 kg/m^2^ and ≥30 kg/m^2^ (Fisher’s exact test). Anorexia (19.8%), followed by diarrhea (10.4%), was the most common gastrointestinal symptom observed. Nausea and diarrhea were more frequent in the SGA-C group than in the SGA-B one (*p* = 0.0491 and *p* = 0.0447, respectively). No differences were found in the other gastrointestinal symptoms between SGA groups.

**Table 2 nutrients-16-00886-t002:** Results of the Subjective Global Assessment.

	Total	SGA-B	SGA-C	*p*-Value
n (%)	96	50 (52.1)	46 (47.9)	-
Weight lost during last 6 months:				
Mean (SD), kg	11.0 (7.1)	7.3 (4.7)	14.9 (7.3)	<0.0001
Mean (SD), %	11.6 (6.7)	8.2 (5.2)	15.3 (6.1)	<0.0001
Physical examination, n (%):				
Subcutaneous fat loss	79 (82.3)	34 (68.0)	45 (97.8)	<0.0001
Loss of muscle mass	80 (83.3)	35 (70.0)	45 (97.8)	0.0002
Malleolar edema	15 (15.6)	7 (14.0)	8 (17.4)	NS
Sacral edema	4 (4.2)	2 (4.3)	2 (4.3)	NS
Ascites	1 (1.0)	0 (0.0)	1 (2.2)	NS
Gastrointestinal symptoms, n (%):				
None	62 (64.6)	37 (74.0)	25 (54.3)	NS
Nausea	4 (4.2)	0 (0.0)	4 (8.7)	0.0491
Vomiting	1 (1.0)	0 (0.0)	1 (2.2)	NS
Diarrhea	10 (10.4)	2 (4.0)	8 (17.4)	0.0447
Dysphagia	4 (4.2)	3 (6.0)	1 (2.2)	NS
Abdominal pain	6 (6.3)	3 (6.0)	3 (6.5)	NS
Anorexia	19 (19.8)	7 (14.0)	12 (26.1)	NS

Abbreviations: SGA-B, moderately malnourished; SGA-C, severely malnourished; NS: not statistically significant.

**Table 3 nutrients-16-00886-t003:** Proportion of moderately or severely malnourished patients according to GLIM criteria.

	Total	Moderate	Severe	*p*-Value
n (%)	96	44 (45.8)	52 (54.2)	-
Weight loss within the past 6 months:				
Mean (SD), kg	11.1 (7.1)	5.9 (3.0)	15.2 (6.9)	<0.0001
Mean (SD), %	11.6 (6.7)	6.3 (2.9)	16.1 (5.6)	<0.0001
Weight loss:				
5–10% within the last six months or 10-20% beyond six months, n (%)	36 (37.5)	34 (77.3)	2 (3.8)	<0.0001
>10% within the last six months or >20% beyond 6 months, n (%)	49 (51.0)	0 (0.0)	49 (94.2)	<0.0001
Low BMI (kg/m^2^): <20 in <70 years or <22 in ≥70 years, n (%)	3 (3.1)	1 (2.3)	2 (3.8)	NS
Reduced muscle mass *:				
Mild to moderate deficit, n (%)	30 (31.3)	20 (45.5)	10 (19.2)	0.0079
Severe deficit, n (%)	18 (18.8)	0 (0.0)	18 (34.6)	<0.0001
Reduced dietary intake (or absorption), n (%)	58 (60.4)	27 (61.4)	31 (59.6)	NS
Inflammation, n (%) **	88 (91.7)	39 (88.6)	49 (94.2)	NS

Abbreviations: BMI, body mass index; GLIM, Global Leadership Initiative on Malnutrition; NS: not statistically significant. * According to validated body composition techniques or anthropometric measurements such as muscle circumference of the arm or the perimeter of the calf, and grip strength as an additional supporting measurement. ** Acute disease/injury or chronic disease-related disease/injury. Disease-related inflammation was identified as a C-reactive protein concentration > 5 mg/L at hospital discharge. The degree of agreement between SGA and GLIM for diagnosing malnutrition was 100%. Nevertheless, the agreement between the different stages of malnutrition (SGA-B, GLIM moderate malnutrition and SGA-C, GLIM severe malnutrition) was moderate (kappa index = 0.502) with a degree of agreement of 75%.

**Table 4 nutrients-16-00886-t004:** Barthel index scores at hospital discharge.

	Total	Men	Women	*p*-Value
n (%)	96 (100.0)	69 (71.9)	27 (28.1)	
BI score, median (IQR)	90 (65–100)	90 (65–100)	85 (70–95)	NS
BI < 100, n (%)	64 (66.7)	42 (60.9)	22 (81.5)	NS
Total dependency	0 (0.0)	0 (0.0)	0 (0.0)	
Severe dependency	8 (8.3)	6 (8.7)	2 (7.4)	
Moderate dependency	20 (20.8)	15 (21.7)	5 (18.5)	
Media dependency	26 (27.1)	15 (21.7)	11 (40.7)	
Minimal dependency	10 (10.4)	6 (8.7)	4 (14.8)	

Abbreviations: BI, Barthel index; IQR, interquartile range; NS: not statistically significant

**Table 5 nutrients-16-00886-t005:** Results of the handgrip strength and TUG tests.

	Total	Men *	Women	*p*-Value
Handgrip strength:				
Mean (SD), kg	21.7 (11.0)	25.0 (10.9)	13.1 (5.0)	<0.0001
<27 men or <16 women; n (%)	60 (62.5)	39 (56.5)	21 (77.8)	NS
TUG test:				
Mean (SD), seconds	20.0 (17.3)	16.7 (14.2)	28.1 (21.5)	0.0004
>20 s, n (%)	26 (27.1)	14 (20.3)	12 (44.4)	0.0224

Abbreviations: TUG, Timed Up-and-Go; NS: not statistically significant; * No dynamometry measures were available for two males.

**Table 6 nutrients-16-00886-t006:** Results of the bioelectrical impedance and nutritional ultrasound assessments.

	Total	Men	Women	*p*-Value
n (%)	96 (100.0)	69 (71.9)	27 (28.1)	
BIA *:				
FFMI, mean (SD) FFMI < 17 men or <15 kg/m^2^ women; n (%)	17.4 (3.9) 33.3	17.7 (4.1) 35.8	16.3 (3.3) 26.1	0.0341 NS
SMMI, mean (SD), kg/m^2^	8.5 (2.4)	8.9 (2.1)	7.5 (2.9)	0.0007
PA, mean (SD), ^o^ PA < 3.95 (%)	4.5 (1.1) 29.5	4.6 (1.1) 30.8	4.4 (0.9) 27.3	NS NS
Nutritional US:				
Subcutaneous abdominal adipose tissue, mean (SD), cm	2.11 (0.9)	1.87 (0.8)	2.72 (0.9)	<0.0001
Preperitoneal adipose tissue, mean (SD), cm	0.9 (0.5)	0.8 (0.4)	1.0 (0.5)	NS
RF-CSA, mean (SD), cm^2^	3.4 (1.3)	3.7 (1.4)	2.6 (0.7)	<0.0001
RF thickness, mean ± (SD), cm	1.0 (0.6)	1.2 (0.5)	0.9 (0.2)	<0.0001
RF circumference, mean (SD), cm	8.7 (1.4)	9.0 (1.4)	7.9 (1.1)	<0.0001
Myosteatosis (%) **	83.7	78.1	100.0	NS

Abbreviations: BIA, bioelectrical impedance analysis; FFMI, fat-free mass index; PA, phase angle; US, ultrasound; RF, rectus femoris; RF-CSA, rectus femoris cross-sectional area; SMMI = skeletal muscle mass index. * Data for FFMI were not available for two males and four females, and data for SMMI were not available for twenty-two males and seven females. ** Percentages were calculated based on sample sizes of 43, 32, and 11 patients for the overall group, men, and women, respectively. Missing data n = 53.

## Data Availability

The raw data supporting the conclusions of this article will be made available by the authors on request.

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
