# Peer review of "Nutritional and Morphofunctional Assessment of Post-ICU Patients with COVID-19 at Hospital Discharge: NutriEcoMuscle Study"

_nutrients, 2024, doi:10.3390/nu16060886_

Round 1

Reviewer 1 Report

Comments and Suggestions for Authors

Dear Authors

Thank you for the opportunity to review this paper. The study presented the weight loss and malnutrition of  COVID patients depending on the hospitalization duration, and the severity of the disease.

The paper is well prepared and the authors used multiple and complex tools to estimate the nutrition status of the patients.

The methodology of the study is very detailed written.

The results of the study are very carefully analysed.

Please find minor comments below:

Line 83 US measurement is the abbreviation of... please write

Line 181 I think is more suitable for writing the selected sample, not the entire population (there are only 96 patients)

Line 201 Characteristics of the sample (not of the population)

Discussion

Maybe another limitation of the study was the use of different BIA devices to assess the body composition.

Kind regards

Author Response

We thank the editor and the reviewers for the effort and time put into the review of our manuscript. We have appreciated very much your constructive comments. Responses to the reviewer and changes in the revised manuscript are as follows.

1- Line 83 US measurement is the abbreviation of... please write

Ultrasound (US).   The word has been included in the mansucript. 

2- Line 181 I think is more suitable for writing the selected sample, not the entire population (there are only 96 patients)

Following the reviewer’s suggestion, the sentence has been modified.

3-Line 201 Characteristics of the sample (not of the population)Ti

Title of table 1 has been modified as suggested.

4- Discussion: Maybe another limitation of the study was the use of different BIA devices to assess the body composition

We agree with the referee that a limitation of our study was the use of different BIA devices to assess the body composition, depending on the hospital where the test was performed.  To minimize bias the total body water, necessary for estimating Fat-Free Mass (FFM), was calculated only by using the 50-kHz impedance. A sentence explaining this limitation has been included in the manuscript. 

Reviewer 2 Report

Comments and Suggestions for Authors

The authors provided results of the NutriEcoMuscle study of COVID-19 patients, which showed that patients were discharged with varying degrees of malnutrition and muscle loss, and lost independence was also noted. The authors also explored ultrasound technology for assessment of body composition. This is a well-conducted study and I have only minor text corrections that I’ve listed in the comments of the attached pdf.

Author Response

We would like to thank the reviewer the thoughtful and thorough review as well as the positive comments on the manuscript.

Following the reviewer’s suggestion, grammatical changes in lines 147 and 363 have been performed.